# Protective Effect of Factor XIII Intron-K G Allele on Subclinical Vascular Disease

**DOI:** 10.3390/ijms262110293

**Published:** 2025-10-22

**Authors:** Barbara Cogoi, Regina Esze, Sándor Somodi, Amir H. Shemirani, Zsuzsanna Bereczky, László Muszbek, György Paragh, Mónika Katkó, Miklós Káplár

**Affiliations:** 1Division of Metabolism, Department of Internal Medicine, Faculty of Medicine, University of Debrecen, 4032 Debrecen, Hungary; cogoi.barbara@belklinika.com (B.C.); esze.regina@med.unideb.hu (R.E.); paragh@belklinika.com (G.P.); 2Department of Emergency Medicine, Faculty of Medicine, University of Debrecen, 4032 Debrecen, Hungary; somodi@med.unideb.hu; 3Division of Clinical Laboratory Science, Department of Laboratory Medicine, Faculty of Medicine, University of Debrecen, 4032 Debrecen, Hungary; amir@med.unideb.hu (A.H.S.); zsbereczky@med.unideb.hu (Z.B.); muszbek@med.unideb.hu (L.M.); 4Division of Endocrinology, Department of Internal Medicine, Faculty of Medicine, University of Debrecen, 4032 Debrecen, Hungary; katko.monika@med.unideb.hu

**Keywords:** FXIII, FXIII polymorphisms, diabetes mellitus, obesity, carotid artery intima–media thickness

## Abstract

Carotid artery intima–media thickness (cIMT), a pre-clinical vascular change that accompanies atherosclerosis is considered as a cardiovascular risk factor. Coagulation factor XIII (FXIII) stabilizes the fibrin clot and increases its resistance to fibrinolysis. Regarding FXIII Val34Leu polymorphism, the protective effect of the Leu34 allele in the presence of elevated fibrinogen levels against myocardial infarction was demonstrated. Our aim was to investigate the effect of FXIII polymorphisms on cIMT. Patients with obesity (*n* = 69), type 2 diabetes mellitus (T2DM) (*n* = 104), and age- and sex-matched healthy controls (*n* = 82) were enrolled. FXIII polymorphisms (Val34Leu, His95Arg, Intron-K C>G) were determined by RT-PCR with FRET detection and melting curve analysis. cIMT was determined by B-mode ultrasound. Differences in cIMT between control (median: 0.5965, IQR: 0.5115–0.6580 mm) and T2DM (median: 0.7105, IQR: 0.5948–0.7568 mm), as well as between obese (median: 0.6105, IQR: 0.5455–0.6780 mm) and diabetic groups, were found (*p* < 0.0001 and *p* = 0.003, respectively). Genotype and allele frequencies of the studied polymorphisms did not differ between subgroups. In the study group (*n* = 255) after adjustment for age and sex, the presence of Intron-K G allele showed a significant and independent protective effect against cIMT progression in a separate model (*p* = 0.005) and after adjusting for other parameters associated with cIMT (*p* = 0.015). FXIII Intron-K G allele provides a protective effect against subclinical vascular disease in the studied population, and this effect is independent of the presence of obesity, as well as T2DM, Leu34 allele, and fibrinogen levels.

## 1. Introduction

Adults with type 2 diabetes (T2DM) have 2 to 4 times higher cardiovascular risk compared to people without this condition [1]. This excess risk cannot be attributed solely to the higher prevalence of well-known Framingham risk factors such as hypertension, dyslipidaemia, and obesity, but genetic factors may also be important [2]. The underlying cause of most CVDs is atherosclerosis, a chronic inflammatory disease of the arterial vessel wall. There is growing evidence that coagulation system in tight connection with inflammatory process plays a pivotal role in the development and the progression of atherosclerosis [3].

Blood coagulation factor XIII (FXIII) is found in both cellular and plasmatic forms. The cellular form (cFXIII) is composed of two A subunits (FXIII-A), and it is present in the cytoplasm of many cell types, like platelets, monocytes, macrophages, chondrocytes, osteocytes, and preadipocytes [4]. The plasmatic form of FXIII (pFXIII) has a different structure. It circulates as a zymogen heterotetramer, composed of two potentially catalytic A subunits and two carrier/inhibitor B subunits (FXIII-B) [4]. FXIII in the plasma plays a pivotal role in the terminal phase of coagulation cascade. Activated FXIII (FXIIIa) stabilizes the clot by crosslinking fibrin gamma and alpha chains, as well as crosslinking fibrin and others, mostly antifibrinolytic proteins like alpha2-antiplasmin (A2AP), plasminogen activator inhibitor 2 (PAI-2), and thrombin-activatable fibrinolysis inhibitor (TAFI) [4,5]. These processes protect the fibrin clot from shear stress and degradation by the fibrinolytic system [4,5,6].

Elevated levels of FXIII were associated with significantly increased risk of MI [7,8,9]. In patients with diabetic angiopathy, FXIII activity and subunit concentrations are significantly increased, and it might contribute to the development of atherosclerotic complications of diabetes [10,11,12].

The gene encoding the FXIII A subunit (F13A) is polymorphic and several studies focused on the role of the genetic variations in cardiovascular and thrombotic diseases [8,9,13,14]. The most investigated polymorphism of the F13A1 is p.Val34Leu (rs5985). It causes earlier activation of FXIII and leads to faster crosslinking of fibrin fibers [4,6,15]. In recent decades, inconsistent results were published regarding the role of p.Val34Leu polymorphism in cardiovascular diseases, but since 2020, several meta-analyses have confirmed the protective effect of the minor Leu34 allele against coronary artery disease (CAD) [13,16,17]. It has been confirmed that Leu34 allele alters the structure of the nascent clot, which becomes more susceptible to fibrinolysis, and results in protection against myocardial infarction [18]. Heterogeneity of the FXIII-B subunit gene (F13B) has also been identified, and the polymorphisms in the carrier subunit seem to have impact on vascular diseases. The p. His95Arg polymorphism (rs6003) is a risk factor for venous thromboembolism and is associated with increased mortality after cerebral ischemia [19,20,21,22].

A known polymorphism in the Intron-K of F13B alters the isoelectric point of the protein and has been supposed to have a profound impact on plasma FXIII levels and disease susceptibility [19,23]. In 2015 a study involving Hungarians demonstrated, that carriers of the Intron-K nt29756 G allele have significantly lower FXIII activity, and antigen levels, and this resulted in significant protection against coronary atherosclerosis (CAS) and MI. A synergistic effect with the FXIII Leu34 allele was also demonstrated, and the combined carriership of the two minor alleles leads to decreased FXIII levels. The same study group proved, that plasma FXIII levels are subjected to multifactorial regulation, in which age, fibrinogen level, and FXIII-B Intron-K polymorphism are major determinants [22,24].

Carotid intima–media thickness (cIMT) is a non-invasive biomarker of the long, subclinical phase of atherosclerotic process, characterized by endothelial dysfunction and gradual thickening of the vessel wall [25,26]. The normal values for an adult range between 650 and 900 µm with an increase of 0–40 µm every year [25]. There is a strong relationship between cIMT and risk factors for atherosclerosis, their duration, and intensity. Mean common carotid artery intima–media thickness independently predicts cardiovascular events, and in 2020, a large-scale meta-analysis involving data from 119 clinical trials showed that interventions reducing cIMT progression are also likely to reduce cardiovascular disease event rates [26,27,28,29].

Atherosclerosis is a progressive, multifactorial, systemic disease that represents the major cause of death and morbidity in patients with diabetes. Risk stratification in this high-risk population is essential to recognize those who are prone to develop early cardiovascular complications.

There is no data regarding the association of carotid IMT with FXIII polymorphisms in patients with diabetes.

The aim of our study was to explore the effect of coagulation factor XIII polymorphisms on carotid artery IMT, as a surrogate measure of generalized vascular disease in obese and type 2 diabetic patients.

## 2. Results

The groups of controls, obese, and type 2 diabetic patients were age- and sex-matched, and their demographic and anthropometric parameters are shown in Table 1. The frequency of obesity (BMI > 30 kg/m^2^) was 79% in the type 2 diabetic group.

Laboratory parameters of glucose and lipid metabolisms, hemostasis factors, as well as C-reactive protein (CRP), glomerular filtration rate (GFR), and mean cIMT value were measured and are shown in Table 2. Higher insulin, C-peptide, and triglyceride levels, as well as lower HDL-C and ApoA1 levels with low-grade inflammation, accompany the obesity and type 2 diabetes, while abnormal HbA1c levels were measured accordingly only in patients with type 2 diabetes. Higher LDL-C levels of obese patients might be explained with the lack of statin treatment in this group. Moderate differences were found in prothrombin time (PT) and thrombin time (TT), PT was higher in type 2 diabetic patients, while TT was higher in obese non-diabetic patients compared to the other studied groups. Mean cIMT was higher in type 2 diabetic group than in controls and obese non-diabetic group.

FXIII polymorphisms (p.Val34Leu, p.His95Arg, Intron-K C>G) were determined in all subjects. Genotype distribution and allele frequencies are shown in Table 3. We found that there is a Hardy–Weinberg equilibrium for p.Val34Leu and Intron-K C>G polymorphisms, while for p.His95Arg polymorphism, the observed genotype distribution differed from the expected genotype distribution except for the subgroup of patients with T2DM.

First, we analyzed whether any of the studied FXIII polymorphisms influenced the subclinical vascular disease progression by comparing mean cIMT values of subjects carrying the minor alleles to subjects with homozygotes for the major alleles. We found in the whole study population that mean cIMT values in carriers of Intron-K G allele were lower than in subject with Intron-K C-C genotype (Figure 1). No such difference was found using different grouping of subjects or in the case of the other examined FXIII polymorphisms.

In the whole study group, after adjustment for age and sex, only the presence of Intron-K G allele among the studied polymorphisms showed significant and independent protective effect against cIMT progression in a separate model (standardized β: −0.156, SE of standardized β: 0.056, *p* = 0.005) or in a model combined with p.Val34Leu and p.His95Leu status (standardized β: −0.160, SE of standardized β: 0.056, *p* = 0.004).

Other studied parameters associated with mean cIMT in the whole population and in subgroups are shown in Table 4. Among categorical variables, male sex and hypertension were associated with higher mean cIMT values. Mean cIMT was higher (*p* = 0.046) in males (median: 0.6540, IQR: 0.5525–0.7465) than in females (median: 0.6150, IQR: 0.5443–0.7065) in the whole study population. Subjects with hypertension had higher mean cIMT values in the whole study population (median: 0.6830, IQR: 0.6000–0.7528 vs. median: 0.5740, IQR: 0.5045–0.6520; *p* < 0.0001), in controls (median: 0.6690, IQR: 0.5980–0.7315 vs. median: 0.5520, IQR: 0.4933–0.6280; *p* = 0.0002), and in obese group (median: 0.6400, IQR: 0.5950–0.7370 vs. median: 0.5760, IQR: 0.5340–0.6465; *p* = 0.01); and mean cIMT was only marginally different in T2DM patients with hypertension (median: 0.7290, IQR: 0.6155–0.7625 vs. median: 0.6643, IQR: 0.5360–0.7420; *p* = 0.057).

Predictors based on the results of a multiple regression analysis in the whole study population are shown in Table 5. The protective effect of Intron-K G allele remained significant in a model including all parameters associated with mean cIMT values.

## 3. Discussion

FXIII is a transglutaminase influencing blood coagulation processes by stabilizing the fibrin clot and increasing its resistance to fibrinolysis. Many studies have tried to clarify its role in the development of cardiovascular diseases.

To our best knowledge, our study is the first to report the association between FXIII polymorphisms and subclinical vascular disease measured by cIMT. We found that the G allele of Intron-K G-C polymorphism has a significant and independent protective effect against arterial injury.

In 2001, Kohler et al. found that elevated FXIII antigen levels were associated with severe coronary sclerosis [30]. In 2007, Bereczky et al. published that FXIII activity and antigen levels in the upper tertile were associated with significantly increased risk of MI in females. They suggested that FXIII activity and antigen levels should be considered as candidate markers in the cardiovascular risk stratification for women [8]. A sex-specific effect of elevated FXIII levels regarding peripheral artery disease (PAD) was described by Shemirani et al. They published that elevated FXIII activity as well as antigen conferred a significantly elevated risk of PAD only to females [31]. Other studies failed to find any association between FXIII p.Val34Leu polymorphism and the risk of PAD [32,33].

The genes of both A and B subunits are polymorphic, and numerous studies have addressed the effect of common polymorphisms on the risk of atherothrombotic diseases [13,14,16,17,18,20,21,22,24,32,33,34,35,36]. The most extensively examined polymorphism is FXIII-A p.Val34Leu, located in the activation peptide. Numerous conflicting results on the association of CAD with this polymorphism support the need for a meta-analysis [37]. In 2007, Vokó et al. reported that the Leu34 allele has a modest protective effect against CAD [16]. In 2014, a meta-analysis by Wang et al., including 12 case–control studies, found that p.Val34Leu polymorphism was associated with significantly decreased MI risk [34]. In 2017, Jung et al. conducted a meta-analysis of 36 studies and confirmed the role of Val allele in increased risk of CAD, but p.Val34Leu polymorphism was not associated with CAD without MI [35]. It was proposed that gene–gene and gene–environmental interaction factors influence the effect of FXIII. Several studies highlight the role of fibrinogen levels, as it alters the effect of Val34Leu polymorphism on the structure of the fibrin clot [13,14,18,38]. The other interaction, which seems to be important, is that with insulin resistance, the cardioprotective effect of the Leu34 allele is lost in the presence of features associated with insulin resistance, particularly in the presence of elevated plasminogen activator inhibitor-1 levels (PAI-1) [36,39]. A strong correlation was demonstrated between FXIII-B and insulin resistance. Increased levels of insulin enhance hepatic synthesis of PAI-1 and insulin possibly has a similar effect on FXIII-B production [36,39,40].

Most of the studies examined the effect of FXIII on MI and not on the overall CAD risk.

The only study investigating the effect of the p.Val34Leu polymorphism on cIMT was the Glostrup study in 2003. They failed to demonstrate the association of the polymorphism either with the cIMT or with the plaque occurrence in the carotid artery [41]. Interestingly, in 2005, a cross-sectional study evaluated whether FXIII had been related to thrombosis and cIMT in primary antiphospholipid syndrome (APS). The FXIII activity was elevated in patients with primary APS, particularly in patients with multiple vessel occlusions and FXIII activity correlated with carotid IMT [42].

Our research group aimed to investigate biomarkers that can be used to identify those patients with diabetes and/or obesity who are at a higher risk of developing vascular complications. Atherosclerosis is partly under genetic control [43,44]. cIMT is considered a marker of arterial injury, which is largely influenced by hereditary factors; therefore, the investigation of the correlations between cIMT and various genetic factors may lead to the identification of new atherosclerosis genes [45]. Our aim was to investigate the effect of three FXIII polymorphisms, namely p.Val34Leu, p.His95Arg, and Intron-K C>G on subclinical vascular disease expressed as cIMT in obesity and patients with type 2 diabetes.

According to our results, cIMT was significantly higher in patients with type 2 diabetes compared to control subjects; furthermore, there was a significant difference in cIMT even between the groups of obese and that of diabetic patients. The minor allele frequencies as well as age, sex, and ratio of smokers did not differ significantly between the control, obese, and diabetes group. After adjusting for age and gender in the whole study group, only the presence of the G allele of the Intron-K polymorphism showed significant and independent protective effect against pre-clinical vascular changes among studied polymorphisms.

The protective effect of the G allele remained significant even after adjustment for obesity, HbA1c, and fibrinogen levels.

The p.Val34Leu polymorphism had no significant effect on cIMT. One possible explanation is that Leu34 allele has effect only on the acute thrombotic process of cardiovascular diseases and not on the subclinical phase of arteriopathy. Another possible explanation may be the high incidence rate of insulin resistance in the entire study group, which destroys the protective effect of the Leu34 allele in cardiovascular diseases. There was no association between p.His95Arg polymorphism and cIMT.

Our results suggest a protective effect of the FXIII-B Intron-K G allele, and recent findings may provide an explanation for this. Muszbek et al. investigated in detail the effects of FXIII on vascular smooth muscle cells and recently published his results [46]. It has already been proven, that FXIII directly enhances the migration and proliferation while inhibiting the apoptosis of endothelial cells, fibroblasts, and monocytes [3,47,48,49]. In the above-mentioned Hungarian study, it was confirmed that vascular smooth muscle cells react with enhanced collagen secretion, proliferation, and migration to activated recombinant FXIII. Sources of activated FXIII in the vessel wall layers may be platelets and macrophages, as well as plasmatic FXIII derived from intraplaque hemorrhage [46]. The G allele of Intron-K polymorphism drastically reduces plasma FXIII levels, so it can be assumed that it is able to reduce the FXIII concentration even within the vessel wall, thereby providing a protective effect.

Examining the effect of polymorphisms separately in the three investigated groups, the effect of the G allele was significant only in the obese group, so it can be said that the protective effect of the minor allele was not diabetes-specific. In the control group and in the diabetic group, the cIMT results of patients with the G allele were also better, but the difference did not reach the level of significance.

Limitations of our study have to be considered. The number of participants enrolled in this research was probably too modest, and this may explain the lack of a significant protective effect of the Intron-K polymorphism in the subgroups studied separately. Also, the case number did not allow us to analyze the effect of minor alleles in homozygotic form. Our study population was entirely Hungarian. Previous studies have suggested that conflicting results regarding the effect of FXIII polymorphisms might be a consequence of gene–gene and gene–environmental interactions. The impact of a genetic condition may vary across different populations due to the varying prevalence of environmental and genetic risk factors. Our Hungarian cohort limits generalizability of our data; however, this population is one of the largest coronary heart disease risk groups in Europe, and the significant impact of a genetic polymorphism on cardiovascular disease may be even greater [36]. We measured cIMT to determine vascular injury, a biomarker that is affected by a number of antidiabetic, antihypertensive, and lipid-lowering medications [50,51]. We cannot exclude the possibility of an unidentified effect of these medications on our results, as we did not record their use.

Our results prove that this polymorphism has a significant influence on arterial wall aging, which is closely related to atherosclerosis. Since FXIII stabilizes the fibrin clot, it is clear how FXIII levels and polymorphisms, which also affect the final clot structure, are related to the acute thrombotic event leading to myocardial necrosis. The role of the coagulation factor in the formation of atherosclerotic plaque, as well as in the thickening of the arterial wall is less well-understood. The protective effect of the FXIII-B Intron-K G polymorphism against CAD can be attributed to its lowering effect of FXIII level. FXIII levels are affected by many factors, like age, smoking, etc., which have a considerable impact on CAD and on cIMT. In our study, the effect of Intron-K G polymorphism remained significant even after adjustment for age, gender, smoking, and hypertension. In our opinion, the minor allele decreases the FXIII concentration within the vessel wall and leads to diminished activation of all cell types involved in the process of atherosclerosis. Since our study does not provide direct evidence, linkage disequilibrium with other functional variants or population-specific effects cannot be ruled out in the background of the observed association.

## 4. Materials and Methods

We recruited participants for the study consecutively from patients attending the Obesitology and Diabetes outpatient clinic at the Department of Internal Medicine of the University of Debrecen. Obesity was defined as a body mass index (BMI) ≥ 30 kg/m^2^. Amongst the patients with type 2 diabetes, those with severe hypoglycemia, hyperglycemia, diabetic ketoacidosis within three months prior to sample collection, active infections, malignancy, and other severe co-morbid illnesses were excluded. We also excluded pregnant females from our study. Only patients aged between 18 and 65 years were included.

Subjects with smoking packyears ≥ 20 were considered as smokers. Criteria for hypertension included two consecutive arterial blood pressure values equal to or exceeding 140/90 mmHg or normal blood pressure while taking antihypertensive medication. Hypertensives were included only if they did not have hypertensive crisis within three months prior to sample collection.

In 2012, altogether, we examined 507 subjects, of which 255 met our inclusion criteria: 69 obese patients, 104 patients with type 2 diabetes, and 82 age- and sex-matched healthy control subjects were involved in our study. We obtained institutional ethical approval and informed consent from participants. The study was approved by the Regional and Institutional Ethical Committee of the University of Debrecen.

Detailed history and socio-demographic data were obtained from all participants. Laboratory analysis and measurement of cIMT was carried out by personnel blinded to the clinical status of the subjects.

Laboratory parameters were measured immediately after sampling at the Department of Laboratory Medicine, University of Debrecen.

Parameters including hemoglobin A1c (HbA1c), insulin, C-peptide, total cholesterol, low-density lipoprotein cholesterol (LDL-C), high-density lipoprotein cholesterol (HDL-C), triglyceride, ApoB100, ApoA1, Lp (a), C-reactive protein (CRP), and fibrinogen serum levels were measured using reagents from Roche (Roche, Basel, Switzerland).

Coagulation parameters included prothrombin time (PT), activated partial thromboplastin time (APTT), and thrombin time (TT) measured by the Siemens BCS XP System (Siemens Healthineers, Erlangen, Germany) using Innovin reagent (Siemens) for PT, Pathromtin SL (Siemens) reagent for APTT and Thrombin LX (Labexpert Ltd., Debrecen, Hungary) for TT. Fibrinogen was measured by the Clauss method using Labexpert reagent. The standardized PT (international normalized ratio—INR) was also reported.

Genomic DNA isolation was executed from peripheral blood leucocytes using QIAamp DNA Blood Mini Kit (Qiagen, Hilden, Germany).

FXIII polymorphisms (p.Val34Leu, p.His95Arg, Intron-K C>G) were determined by real-time PCR with fluorescence resonance energy transfer detection and melting curve analysis [22,52].

Philips HD 11 XE ultrasound equipment with a 7.5 MHz linear transducer was used to measure IMT (mm). Online measurements of IMT were performed in the far artery wall of the common carotid arteries, 10 mm proximal to the carotid bulb. All measurements were performed on frozen, enlarged images at end-diastole, and the transducer was in the medio-lateral direction. IMT was measured on a 1 cm segment. In each of these 1 cm segments, 10 measurements of IMT were performed at 1 mm increments on both sides. The mean IMT of the 20 values in each patient was calculated [53].

Statistical analysis was performed by STATISTICA 14 software (Statsoft Inc., Tulsa, OK, USA). The distribution of continuous variables was checked by the Kolmogorov–Smirnov test. Multiple group comparisons were performed using Kruskal–Wallis H test. The stochastic relationships of discrete variables and testing allele frequencies for Hardy–Weinberg equilibrium were analyzed by Chi-square test. Pearson’s correlations were performed to test the relationships between continuous variables (variables were logarithmically transformed in case of non-normal distribution). Multiple linear regression analysis was performed using logarithmically transformed mean cIMT as a dependent variable, and age, sex (coded as 0 = female, 1 = male), and the carrier status regarding each FXIII polymorphism (coded as 0 = wild type, 1 = carrier of the recessive allele), as well as, in other models, hypertension (coded as 0 = no hypertension, 1 = hypertension) and logarithmically transformed continuous variables associated with cIMT values, as potential predictor variables. *p* values below 0.05 were considered statistically significant.

## 5. Conclusions

Our study is the first to demonstrate a significant and independent effect of FXIII-B Intron-K G polymorphism on arterial wall thickening. The presence of the minor G allele was associated with decreased cIMT in a population including non-obese, obese, and T2DM patients, and its protective effect was independent of obesity, gender, age, as well as HbA1C and fibrinogen levels. Further research is needed to understand the exact role of this polymorphism in the process of atherosclerosis, and new studies are needed to confirm that the FXIII-B Intron-K G polymorphism can be used as a biomarker to determine susceptibility to increased atherosclerosis.

## Figures and Tables

**Figure 1 ijms-26-10293-f001:**
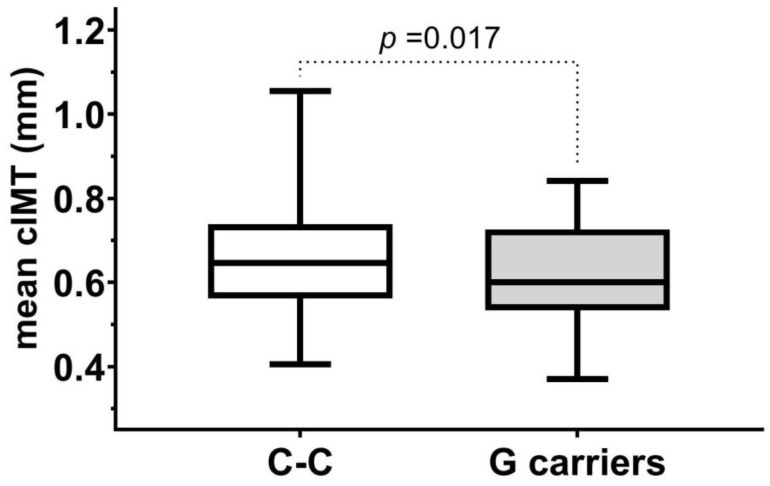
Mean cIMT in subjects with C-C genotypes and C-G or G-G genotypes (G carriers) of Intron-K C>G polymorphism in the study population.

**Table 1 ijms-26-10293-t001:** Demographic and anthropometric data of study populations.

Parameter	Controls *n* = 82	Obese Patients *n* = 69	T2DM Patients *n* = 104	*p* *
Female/male	45/37 (45% male)	38/31 (45% male)	53/51 (49% male)	0.820
Age (years)	52 (44–57) [31–64]	52 (41–58) [32–65]	53 (46–59) [31–65]	0.413
BMI (kg/m^2^)	25.0 (22.9–26.6) [17.9–29.4]	34.6 (31.3–38.4) [30.0–55.3]	34.3 (30.6–37.6) [21.0–52.8]	C vs. Ob < 0.0001 C vs. T2DM < 0.0001
Waist circumference (cm)	96 (88–102) [72–140]	112 (104–122) [82–156]	114 (108–122) [78–170]	C vs. Ob < 0.0001 C vs. T2DM < 0.0001
Smoking (yes/no, *n*)	14/55, *n* = 69 (20% smoker)	8/51, *n* = 59 (14% smoker)	17/69, *n* = 86 (20% smoker)	0.550

Continuous variables are shown as median (IQR) [minimum–maximum]. * *p* values are the results of Chi-square test for female/male ratio and Kruskal–Wallis H test for continuous variables with post hoc test in case of significant results.

**Table 2 ijms-26-10293-t002:** Laboratory parameters and mean cIMT values of study groups.

Parameters	Controls *n* = 82	Obese Patients *n* = 69	T2DM Patients *n* = 104	*p* *
HbA1c (%)	5.1 (5.4–5.8) [4.6–6.5]	5.6 (5.3–5.9) [4.7–6.5]	7.5 (6.7–8.9) [5.3–13.8]	C vs. T2DM < 0.0001 Ob vs. T2DM < 0.0001
Insulin (mU/L)	7.1 (4.6–10.0) [2.1–42.2]	12.7 (8.7–17.2) [3.6–41.7]	13.0 (7.5–20.9) [2.1–66.2]	C vs. Ob < 0.0001 C vs. T2DM < 0.0001
C-peptide (pmol/L)	584 (503–752) [230–2068]	932 (736–1198) [415–2725]	911 (565–1322) [69–2212]	C vs. Ob < 0.0001 C vs. T2DM < 0.0001
Total cholesterol (mmol/L)	5.1 (4.6–5.8) [3.4–7.2]	5.3 (4.7–6.1) [3.1–8.9]	4.9 (4.1–6.0) [3.1–11.0]	0.159
LDL-cholesterol (mmol/L)	3.1 (2.6–3.7) [1.4–5.3]	3.4 (2.8–4.2) [1.3–5.6]	2.9 (2.2–3.8) [1.3–8.5]	Ob vs. T2DM 0.006
HDL-cholesterol (mmol/L)	1.7 (1.4–1.9) [0.9–3.0]	1.3 (1.0–1.7) [0.7–4.1]	1.2 (1.0–1.5) [0.6–2.7]	C vs. Ob < 0.0001 C vs. T2DM < 0.0001
Triglyceride (mmol/L)	1.1 (0.7–1.5) [0.4–4.5]	1.5 (1.1–1.9) [0.5–13.1]	2.0 (1.3–2.8) [0.4–14.1]	C vs. Ob < 0.001 C vs. T2DM < 0.0001 Ob vs. T2DM 0.016
ApoA1 (g/L)	1.76 (1.55–1.95) [1.06–2.61]	1.58 (1.36–1.82) [1.01–2.57]	1.47 (1.29–1.70) [0.94–2.33]	C vs. Ob < 0.005 C vs. T2DM < 0.0001
ApoB100 (g/L)	0.95 (0.83–1.08) [0.47–1.65]	1.08 (0.91–1.28) [0.56–1.68]	1.02 (0.83–1.16) [0.40–2.38]	C vs. T2DM 0.007
Lp (a) (mg/L)	69 (<30–268) [<30–1110]	56 (<30–224) [<30–1572]	135 (54–447) [<30–1919]	C vs. T2DM 0.026 Ob vs. T2DM 0.005
PT (s)	7.8 (7.6–8.1) [7.2–9.4]	7.8 (7.6–8.0) [7.2–20.6]	8.1 (7.8–8.6) [7.2–29.0]	C vs. T2DM 0.0001 Ob vs. T2DM < 0.0005
INR	0.93 (0.91–0.96) [0.85–1.10]	0.93 (0.91–0.96) [0.87–2.16]	0.93 (0.91–0.99) [0.85–2.98]	0.790
APTT (s)	27.8 (26.4–29.7) [21.5–34.3]	27.2 (25.7–28.6) [21.1–46.0]	26.8 (25.2–29.4) [20.5–47.7]	0.185
TT (s)	16.6 (15.7–17.6) [14.2–19.5]	17.5 (16.7–18.2) [14.4–19.5]	16.0 (15.1–17.1) [13.4–21.9]	C vs. Ob 0.002 Ob vs. T2DM < 0.0001
Fibrinogen (g/L)	3.72 (3.17–4.42) [2.04–7.8]	3.66 (3.40–4.38) [2.57–6.75]	3.94 (3.38–4.61) [2.69–7.34]	0.078
CRP (mg/L)	1.1 (<0.5–2.4) [<0.5–73.8]	3.1 (1.1–6.3) [<0.5–25.8]	3.7 (1.9–8.0) [<0.5–39.8]	C vs. Ob < 0.0001 C vs. T2DM < 0.0001
GFR (ml/min/1.73 m^2^) (>90/60–90/30–60)	51/31/0	43/22/4	76/26/2	0.146
Mean cIMT (mm)	0.5965 (0.5115–0.6580) [0.4005–0.8765]	0.6105 (0.5455–0.6780) [0.4170–0.9145]	0.7105 (0.5948–0.7568) [0.3700–1.0550]	C vs. T2DM < 0.0001 Ob vs. T2DM 0.003

Continuous variables are shown as median (IQR) [minimum–maximum]. * *p* values are the results of Chi-square test for GFR categories and Kruskal–Wallis H test for continuous variables with post hoc test in case of significant results.

**Table 3 ijms-26-10293-t003:** Genotype and allele frequencies of the studied FXIII polymorphisms in the total study population and by subgroups.

	Total *n* = 255	Controls *n* = 82	Obese Patients *n* = 69	T2DM Patients *n* = 104	*p*
Val34Leu					
Val/Val	130	36	37	57	0.368
Val/Leu	102	37	24	41	
Leu/Leu	23	9	8	6	
Leu allele frequency	29%	34%	29%	25%	0.236
HWE *p* value	0.825	>0.999	0.509	0.877	
His95Arg					
His/His	196	63	51	82	0.492
His/Arg	44	12	13	19	
Arg/Arg	15	7	5	3	
Arg allele frequency	15%	16%	17%	12%	0.407
HWE *p* value	<0.0001	0.0002	0.037	0.631	
Intron-K C>G					
C/C	188	65	48	75	0.489
C/G	66	17	21	28	
G/G	1	0	0	1	
G allele frequency	13%	10%	15%	14%	0.389
HWE *p* value	0.206	0.527	0.275	0.716	

**Table 4 ijms-26-10293-t004:** Parameters associated with mean cIMT values.

Explanatory Variables	r	*p*
Age	0.420	<0.0001
BMI	0.258	0.0001
Waist circumference	0.278	<0.0001
CRP	0.180	0.004
HbA1c	0.284	<0.0001
C-peptide	0.148	0.020
APTT	0.143	0.023
Fibrinogen	0.153	0.016
Lp(a)	0.180	0.004
Triglyceride	0.227	0.0003
ApoA1	−0.170	0.007
HDL-C	−0.214	0.0006

**Table 5 ijms-26-10293-t005:** Predictors of mean cIMT values in the whole study population.

Dependent Variable: Mean cIMT			
Predictors	Standardized β	SE of Standardized β	*p* Value
Age	0.399	0.066	<0.0001
Male sex	0.165	0.074	0.026
HbA1c	0.185	0.077	0.017
Hypertension	0.162	0.069	0.020
Intron-K G carrier	−0.146	0.059	0.015

## Data Availability

The original contributions presented in this study are included in the article. Further inquiries can be directed to the corresponding author.

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
