# Peer review of "Protective Effect of Factor XIII Intron-K G Allele on Subclinical Vascular Disease"

_ijms, 2025, doi:10.3390/ijms262110293_

Round 1

Reviewer 1 Report

Comments and Suggestions for Authors

Manuscript ID: ijms-3892723

Comments to authors:

In the current manuscript Cogoi et al. investigated the effect of blood coagulation factor XIII (FXIII) polymorphism on carotid artery intima-media thickness and showed that FXIII Intron-K G allele has protective effects against subclinical vascular disease. The paper represents an important field of research that demonstrates a link between changes in FXIII and the risk of cardiovascular disease. The manuscript is logically organized and structured, and methods are used as appropriate, however, several comments should be taken into consideration in order to improve manuscript quality (presented below).

  1. Please provide reference to support the statement in the next lines: 38-39; 48-50; 50-52; 52-54; 55-58; 59-62; 64; 72-73; 73-75; 93-94; 95-96; 101-102; 104-107; 107-108; 112-114; 114-115; 224-225.
  2. Table 1: can authors additionally provide a number of patients but not only % (smoking patients number out of the whole number) in the “smoking” category?
  3. Should methods section be listed after introduction? Please correct it.
  4. Line 373-374: this statement is not required: “For research articles with several authors, a short paragraph specifying their individual contributions must be provided”.
  5. Line 374: this statement is not required: “The following statements should be used”

Author Response

We appreciate the Reviewer’s helpful suggestions. We have provided detailed answers to all comments, and the altered text has been indicated in the revised version of the manuscript. Please find here our point-by-point response.

Comments to authors:

In the current manuscript Cogoi et al. investigated the effect of blood coagulation factor XIII (FXIII) polymorphism on carotid artery intima-media thickness and showed that FXIII Intron-K G allele has protective effects against subclinical vascular disease. The paper represents an important field of research that demonstrates a link between changes in FXIII and the risk of cardiovascular disease. The manuscript is logically organized and structured, and methods are used as appropriate, however, several comments should be taken into consideration in order to improve manuscript quality (presented below).

Please provide reference to support the statement in the next lines:

38-39: According to the other Reviewer’s suggestion we shortened the Introduction section. This sentence has been removed.

48-50; 50-52; 52-54; 55-58; 59-62: Relevant citations have been inserted after these sentences instead of after the paragraph. (lines 49-65)

64: According to the other Reviewer’s suggestion we shortened the Introduction section. This sentence has been removed.

 72-73; 73-75; 93-94; 95-96; 101-102; 104-107; 107-108: This part of the Introduction has been revised, and references have been added accordingly. (lines 71-124)

112-114; 114-115: Relevant citations have been inserted after these sentences instead of after the paragraph. (lines 125-128)

224-225: Relevant references have been added to this sentence. (lines 239-241)

Table 1: can authors additionally provide a number of patients but not only % (smoking patients number out of the whole number) in the “smoking” category?

There are some missing data regarding participants’ smoking habits. Table 1 has been completed with the exact number of patients with different smoking habits.

Should methods section be listed after introduction? Please correct it.

The order of the sections is determined by the journal, we followed the guideline.

Line 373-374: this statement is not required: “For research articles with several authors, a short paragraph specifying their individual contributions must be provided”.

Line 374: this statement is not required: “The following statements should be used”

Thank you for pointing this out. We have deleted the unnecessary text. (lines 407-408)

Reviewer 2 Report

Comments and Suggestions for Authors

This manuscript presents a clinical study investigating the association between coagulation factor XIII polymorphisms (Val34Leu, His95Arg, and Intron-K C>G) and carotid artery intima-media thickness (cIMT) as a surrogate marker of subclinical atherosclerosis in obese and type 2 diabetes patients. The authors report that the FXIII-B Intron-K G allele has a significant and independent protective effect against cIMT progression.

The study addresses an important and timely question linking coagulation genetics with vascular disease risk stratification. The manuscript is well-written, structured, and mostly clear. The findings are novel, as there are no prior data on FXIII polymorphisms and cIMT in diabetes or obesity.

I have some minor comments to improve the manuscript.

1) The introduction is very detailed but overly long; it could be shortened to highlight the study’s novelty more directly.

2) The study population is entirely Hungarian. While this is a valid cohort, genetic effects may vary by ethnicity. The lack of external validation limits generalizability. Please comment extensively on this issue in the discussion section.

3) Although major risk factors were adjusted for, it is unclear whether statin use, antihypertensives, or lifestyle factors (diet, physical activity) were considered. These could strongly affect cIMT. Please comment extensively on this issue in the discussion section.

Author Response

Thank you for taking the time to review our manuscript and for your comments. We have performed changes to the manuscript according to your suggestions and the altered text has been indicated in the revised version of the manuscript. Please find here our point-by-point response.

Comments and Suggestions for Authors:

This manuscript presents a clinical study investigating the association between coagulation factor XIII polymorphisms (Val34Leu, His95Arg, and Intron-K C>G) and carotid artery intima-media thickness (cIMT) as a surrogate marker of subclinical atherosclerosis in obese and type 2 diabetes patients. The authors report that the FXIII-B Intron-K G allele has a significant and independent protective effect against cIMT progression.

The study addresses an important and timely question linking coagulation genetics with vascular disease risk stratification. The manuscript is well-written, structured, and mostly clear. The findings are novel, as there are no prior data on FXIII polymorphisms and cIMT in diabetes or obesity.

I have some minor comments to improve the manuscript.

  • The introduction is very detailed but overly long; it could be shortened to highlight the study’s novelty more directly.

Thank you for your suggestion. The Introduction section has been shortened. (lines 37-117)

  • The study population is entirely Hungarian. While this is a valid cohort, genetic effects may vary by ethnicity. The lack of external validation limits generalizability. Please comment extensively on this issue in the discussion section.

We agree with this, and we discussed this issue in the revised version of the manuscript in the Discussion section. (lines 312-319)

  • Although major risk factors were adjusted for, it is unclear whether statin use, antihypertensives, or lifestyle factors (diet, physical activity) were considered. These could strongly affect cIMT. Please comment extensively on this issue in the discussion section.

Although we agree with the Reviewer that medication and lifestyle factors contribute to the progression of cIMT we cannot calculate with these factors due to lack of information. This limitation has been inserted into the Discussion section. (lines 319-322)

Reviewer 3 Report

Comments and Suggestions for Authors

This manuscript explores the association between factor XIII (FXIII) polymorphisms and subclinical atherosclerotic changes assessed by carotid intima-media thickness (cIMT) in patients with obesity, type 2 diabetes mellitus (T2DM), and healthy controls. The authors identify a potentially protective role of the FXIII-B Intron-K G allele against cIMT progression, independent of traditional risk factors and other FXIII variants. The study addresses a mechanistically interesting question linking coagulation genetics with vascular remodeling.

However, while the hypothesis is biologically plausible and the methodology adequate, the manuscript requires substantial improvement in focus, structure, and scientific clarity. This reviewer has several comments as outlined below.

Major Comments:

  1. Introduction section. The introduction is overly long and partly repetitive, diluting the central message. Should be more concise and shortened.
  2. Sample size. The sample size (n=255) is modest, with subgroup analyses further reducing statistical power. The study includes only 69 obese and 104 T2DM subjects; stratified analyses may therefore be underpowered. This should be added to the Limitation section.
  3. Inclusion criteria. The inclusion/exclusion criteria are clearly described, but it is unclear whether participants were recruited consecutively or selectively. Please clarify the recruitment process and potential selection bias.
  4. The conclusion that the Intron-K G allele is protective is intriguing but may overstate causality. The authors should acknowledge that the observed association may reflect linkage disequilibrium with other functional variants or population-specific effects.

Minor comment:

  1. Ethics approved number. The authors should include the approved number from the Ethical Committee of University of Debrecen.

Author Response

Thank you for your time and constructive comments. We detail below how the manuscript has been complemented according to your suggestions, and the altered text has been indicated in the revised version of the manuscript. Please find here our point-by-point response.

Comments and Suggestions for Authors

This manuscript explores the association between factor XIII (FXIII) polymorphisms and subclinical atherosclerotic changes assessed by carotid intima-media thickness (cIMT) in patients with obesity, type 2 diabetes mellitus (T2DM), and healthy controls. The authors identify a potentially protective role of the FXIII-B Intron-K G allele against cIMT progression, independent of traditional risk factors and other FXIII variants. The study addresses a mechanistically interesting question linking coagulation genetics with vascular remodeling.

However, while the hypothesis is biologically plausible and the methodology adequate, the manuscript requires substantial improvement in focus, structure, and scientific clarity. This reviewer has several comments as outlined below.

Major Comments:

  1. Introduction section. The introduction is overly long and partly repetitive, diluting the central message. Should be more concise and shortened.

The Introduction section has been shortened and revised in accordance with your suggestion. (lines 37-117)

  1. Sample size. The sample size (n=255) is modest, with subgroup analyses further reducing statistical power. The study includes only 69 obese and 104 T2DM subjects; stratified analyses may therefore be underpowered. This should be added to the Limitation section.

We agree with the Reviewer, limitations due to the modest sample size were discussed in the revised manuscript in the Discussion section. (lines 308-312)

  1. Inclusion criteria. The inclusion/exclusion criteria are clearly described, but it is unclear whether participants were recruited consecutively or selectively. Please clarify the recruitment process and potential selection bias.

Thank you for pointing this out. Participants were recruited consecutively to minimize selection bias. It has been clarified in the revised version of the manuscript in the Materials and Methods section. (line 339)

  1. The conclusion that the Intron-K G allele is protective is intriguing but may overstate causality. The authors should acknowledge that the observed association may reflect linkage disequilibrium with other functional variants or population-specific effects.

Thank you for drawing attention to this, we completely agree, a sentence acknowledging this option has been inserted to the end of the Discussion section. (lines 334-337)

Minor comment:

  1. Ethics approved number. The authors should include the approved number from the Ethical Committee of University of Debrecen.

The date and the registration number of the approval have been inserted into the Institutional Review Board Statement. (lines 420-421)

Round 2

Reviewer 3 Report

Comments and Suggestions for Authors

This reviewer has no further comment.